# Tumor Microenvironment in Glioma Invasion

**DOI:** 10.3390/brainsci12040505

**Published:** 2022-04-15

**Authors:** Sho Tamai, Toshiya Ichinose, Taishi Tsutsui, Shingo Tanaka, Farida Garaeva, Hemragul Sabit, Mitsutoshi Nakada

**Affiliations:** Department of Neurosurgery, Faculty of Medicine, Institute of Medical, Pharmaceutical and Health Sciences, Kanazawa University, Kanazawa 920-8641, Japan; sho.tamai@med.kanazawa-u.ac.jp (S.T.); t-ichinose@med.kanazawa-u.ac.jp (T.I.); t.taishi@med.kanazawa-u.ac.jp (T.T.); t-shingo@med.kanazawa-u.ac.jp (S.T.); f.garaewa@mail.ru (F.G.); amuras@med.kanazawa-u.ac.jp (H.S.)

**Keywords:** glioma, microenvironment, invasion

## Abstract

A major malignant trait of gliomas is their remarkable infiltration capacity. When glioma develops, the tumor cells have already reached the distant part. Therefore, complete removal of the glioma is impossible. Recently, research on the involvement of the tumor microenvironment in glioma invasion has advanced. Local hypoxia triggers cell migration as an environmental factor. The transcription factor hypoxia-inducible factor (HIF) -1α, produced in tumor cells under hypoxia, promotes the transcription of various invasion related molecules. The extracellular matrix surrounding tumors is degraded by proteases secreted by tumor cells and simultaneously replaced by an extracellular matrix that promotes infiltration. Astrocytes and microglia become tumor-associated astrocytes and glioma-associated macrophages/microglia, respectively, in relation to tumor cells. These cells also promote glioma invasion. Interactions between glioma cells actively promote infiltration of each other. Surgery, chemotherapy, and radiation therapy transform the microenvironment, allowing glioma cells to invade. These findings indicate that the tumor microenvironment may be a target for glioma invasion. On the other hand, because the living body actively promotes tumor infiltration in response to the tumor, it is necessary to reconsider whether the invasion itself is friend or foe to the brain.

## 1. Introduction

Glioma is one of the most common primary brain tumors, accounting for approximately 25% of central nervous system tumors [1]. Current treatment options for gliomas are surgical resection and adjuvant chemoradiotherapy if they are classified as malignant [2]. The prognosis for patients with gliomas is poor. The median overall survival of patients with glioblastoma (GBM), which is classified as grade 4 by the World Health Organization, is only 8 to 18 months [1,2,3]. One of the reasons for the poor prognosis of gliomas is their invasive nature. Glioma cells diffusely invade the remote region through the normal brain tissue [4]. Furthermore, previous studies have reported that alteration of the surrounding microenvironments contributes to their invasiveness.

Although there have been many in vitro and in vivo studies on glioma, some of the mechanisms underlying glioma invasion remain unclear. One of the major reasons for this is the complex system of glioma invasion [5,6]. First, disruption of intercellular connections occurs. Subsequently, glioma cells attach to the extracellular matrix (ECM). The next step in invasion is ECM remodeling, which involves the destruction and production of ECM proteins. In each step, many factors, molecules, and pathways are involved. Thus, a large number of signaling pathways and molecules are involved in glioma invasion. Therefore, elucidating the mechanisms that underlie glioma invasion is difficult. Another reason is the difficulty of establishing an appropriate glioma invasion model. Some xenograft models using glioma cell lines have been widely used. However, these brain tumors show a clear boundary from the normal brain tissue [7], which is different from real GBM. Glioma is composed of heterogeneous cell populations, some of which are glioma initiating cells (GICs), which have the ability to self-renew and differentiate [8,9]. Previous studies have revealed that GICs play an important role in glioma invasion, and xenograft models using GICs, which were established from patient tumor tissue, sustained their invasive nature [10,11,12].

In this review, we focused on the invasion mechanisms associated with changes in the microenvironment surrounding glioma and categorized these factors as follows: the physical environment, interaction with surrounding cells, and alteration of the microenvironment affected by therapies.

## 2. Physical Environment

### 2.1. Hypoxia

Hypoxia is very common in gliomas. A series of studies has been devoted to measuring the oxygen concentration in glioma tissues compared to that in the normal brain [13,14,15]. The major causes of hypoxia are increased oxygen consumption by actively proliferating glioma cells and endothelial damage [16,17,18]. Hypoxia is one of the most potent triggers for glioma cell invasion [17]. This section discusses the mechanisms by which glioma cells activate invasiveness in hypoxic environments.

#### 2.1.1. Signaling Pathway Associated with Hypoxia-Inducible Factor (HIF)

HIFs are essential transcription factors that regulate responses to hypoxia and play a key role in cancer cell proliferation, invasion, and treatment resistance, thereby supporting the phenotype of cancer stem cells [19,20]. The subtypes of HIF expressed in the brain tissue are HIF-1 and HIF-2, and previous studies have revealed the mechanism of HIF-1. HIF is a heterodimeric complex consisting of two parts: subunit α (regulated by O_2_) and subunit β (constitutively expressed). Under normoxic conditions, prolyl hydroxylases hydroxylate two proline residues of HIF-1α, leading to proteasomal degradation of HIF-1α [21]. HIF-1β is present in cells and can dimerize with other transcription factors. Therefore, HIF-1α determines HIF activity [22]. In hypoxic condition, the complex of HIF-1α and HIF-1β mediates an adaptive transcriptional response to hypoxia, including glycolytic metabolism activation, pro-angiogenic factor secretion, genetic instability in tumor cells, and increased cell migration [23,24,25]. Therefore, HIFs are key molecules that contribute to the invasion of glioma cells in several steps (Figure 1).

One of the major factors involved in the disruption of intercellular connections is zinc finger E-box binding homeobox 1 (ZEB1), which affects cadherins, the main class of cell adhesion molecules in glioma cells [26]. ZEB1 is a transcription factor that contains two zinc finger clusters and is activated by HIF1. ZEB1 plays an important role in glioma invasion [27], and the expression level of ZEB1 is associated with poor prognosis in patients with GBM [26,28]. Analysis of xenograft mouse brain tumor models revealed that knockdown of ZEB in glioma cells leads to a less invasive phenotype of tumor cells [26,28]. ZEB1, via the Roundabout Guidance Receptor 1 (ROBO1) protein, destroys N-cadherin’s anchorage to the cytoskeleton and leads to a highly invasive phenotype [29,30]. ZEB1 also induces a proneural mesenchymal transition (PMT) [26]. Another important molecule is integrin, which is a heterodimer consisting of two subunits, α and β chains, and there are at least 18 types of α subunits and eight types of β subunits. Integrins form focal complexes and contacts connecting the cytoskeleton and extracellular binding sites and subserve signaling to regulate ligand binding affinity [31,32]. Integrins also regulate the stabilization of HIF-1α through the focal adhesion kinase (FAK), Ras homolog family member B (RhoB), and glycogen synthase kinase 3β (GSK-3β) pathways. Hypoxia enhances the expression of αvβ3 and αvβ5 integrins on the cell membrane by activating FAK, and FAK activation leads to an increase in the expression levels of RhoB [33,34]. By reducing the level of Akt, RhoB increased the expression level of GSK-3β in cells while impairing the stabilization of HIF-1α. Activated Akt inversely regulates GSK-3β and prevents HIF-1α from being recognized as a substrate, thereby contributing to the stabilization of HIF-1α [33]. During ECM remodeling, matrix metalloproteinases (MMPs), which are zinc-dependent proteins, play an important role in the destruction of ECM foundations [35]. Silencing HIF-1α results in reduced MMP2 expression and sharply decreased MMP2 and MMP9 enzyme activities [36]. Therefore, HIF activity is associated with MMP activation.

#### 2.1.2. GIC-Based Mechanism

There are two major mechanisms of invasion related to GICs under hypoxia: stimulating stemness, and PMT. Some molecules contribute to the stimulation of glioma stemness in hypoxic environments. Ten–eleven translocation 1 and 3 demethylases activated in hypoxia regulate major pluripotency factors, such as octamer-binding transcription factor 4, Nanog, and signal transducer and activator of transcription 3 (STAT3) [37]. Overexpression of pluripotency factors sustains the stemness of GICs in hypoxic environments and contributes to invasion [37].

PMT is the major mechanism contributing to the increased invasiveness of GICs. Two ways to improve hypoxic conditions in GICs have been identified: the stimulation of angiogenesis, and invasion of normoxic parenchyma [38]. Each mechanism is associated with GIC subtypes. GICs can be classified into three subtypes: classical, mesenchymal, and proneural. Each subtype indicates different properties. The classical subtype contributes to escaping apoptosis [39]. The mesenchymal subtype is associated with the stimulation of angiogenesis, and the proneural subtype subserves in invasion [40]. Under normoxic conditions, GICs shift to the mesenchymal subtype with the upregulation of HIF, STAT3, and basic helix–loop–helix family member 40 and tend to angiogenic tumors by stimulating vascular endothelial growth factor (VEGF) [40]. In contrast, GICs shift to the proneural subtype under hypoxic conditions and stimulate invasiveness [38].

#### 2.1.3. Changing of Intracellular Metabolic Systems

There are two major parallel pathways for glucose metabolism in glioma cells, namely the pentose phosphate pathway (PPP), and glycolysis. In hypoxia, PPP-associated enzymes are downregulated, and glycolysis is stimulated in GICs [41]. In particular, aldolase, a key enzyme in the glycolysis pathway, is strongly upregulated under hypoxic conditions, and lactate, which contributes to the acidification of the surrounding microenvironment, is metabolized [42]. Regarding the interaction of these mechanisms, changes in the intracellular metabolic system enhance hypoxia and induce glioma cell invasion. Thus, each stage of glioma cell invasion is mediated by hypoxia.

### 2.2. Extracellular Matrix

Many types of ECMs surround both normal brain and glioma tissues. Originally, the ECMs subserve essential processes, such as embryonic development, tissue repair, and inflammation in normal tissues [35,43]. However, previous reports have revealed that the interactions between ECMs and glioma cells are associated with glioma migration and invasion. Herein, we focused on the contribution of ECMs to the microenvironment of gliomas and their invasion (Figure 2).

#### 2.2.1. Normal Brain ECM

The major subtypes of macromolecules in normal brain ECM are structural glycoproteins (such as collagen, elastin, fibronectin, and laminin), proteoglycans (such as heparan sulfate proteoglycans), and glycosaminoglycans (such as hyaluronic acid (HA)), which are intricately mixed and have a unique composition [44]. A well-known function of normal brain ECM is that structural glycoproteins regulate the migration of neurons in several regions of the developing and adult brain [45]. The function of the ECM in the normal brain is not only related to tissue structure, but ECM proteins specifically bind to cell surface receptors and activate signaling cascades to regulate neuronal structure and function [46]. Thus, it has become clear that the ECM plays a huge role in promoting cell migration and invasion, determining tumor cell fate, cell maturation and differentiation, cell survival, and tissue homeostasis, and driving tumor progression [47,48,49].

#### 2.2.2. ECM Promoting Invasion

The ECM surrounding brain tumors shares many components with the ECM of the normal adult brain, and most of these are involved in the behavior of glioma cells [50]. Glioma cells also secrete their own ECM components such as HA, brevican, tenascin-C, fibronectin, and thrombospondin, which enhance the mobility and invasiveness of glioma cells [51]. Therefore, by interacting with the ECM, glioma cells can actively migrate using the blood vessels and axons as guide paths [52]. In addition, specific integrins and other receptors involved in tumor cell–ECM component interactions are expressed in glioma cells, and they further promote tumor cell migration and invasion. Integrin α3 has been reported to be involved in the invasion of GICs and to act through the extracellular signal-regulated kinase (ERK) 1/2 pathway [53]. Moreover, α3β1 integrins, which are reported to be highly expressed in gliomas, are also known to play an important role in glioma cell motility and invasion [54]. Integrin α5 is known to interact with fibronectin to promote cell invasion via FAK and has an important mechanism that promotes cell–cell adhesion in fibronectin-poor microenvironments such as the normal brain and promotes cell dispersion in fibronectin-rich microenvironments such as glioma tissue [43,55]. Another study revealed that fibrin-3, an extracellular protein released from GBM cells, also drives oncogenic NF-κB in tumor cells by activating disintegrin and metalloproteinase (ADAM) 17 [56]. So that, these ECMs promote invasiveness of glioma cells by altering cellular signaling pathways. Another important factor related to glioma invasion is rigidity. The mechanical rigidity of ECM positively regulates glioma cell proliferation and invasion [57]. The ECM is remodeled, and its rigidity is significantly increased compared to both juvenile and mature states during cancer genesis [51]. It has been reported that ECM stiffness correlate with glioma grading, and associate with tumor invasiveness and poor prognosis [58].

#### 2.2.3. ECM Inhibiting Invasion

Testican-3 has been reported as an ECM that inhibits glioma invasion. It inhibits tumor invasion by inhibiting membrane-type MMPs (MT-MMPs). However, it is downregulated in glioma tissue and is therefore considered insufficient for invasion inhibition [59]. It has also been reported that fibronectin and fibronectin matrix assembly (FNMA) inhibits the motility and invasive ability of GBM cells [60,61]. The activation of FNMA markedly increases the strength of cell–cell adhesion and cell–cell adhesion to the ECM, resulting in a decrease in tumor invasiveness.

#### 2.2.4. Enzymes Associated with ECM Foundation and Glioma Invasion

Proteolysis is one of the most important factors that affect ECM interactions. The normal brain ECM functions as a barrier to glioma cell invasion. However, invasive glioma cells express ECM-degrading proteases, such as MMPs, ADAMs, urokinase-type plasminogen activator (uPA), and cathepsin B; invade tissue through complex proteolysis; and remodel the ECM microenvironment [5] (Figure 2). One of the major enzymes in the MMP family is a secretory-independent endopeptidase that catalyzes the hydrolytic activities of extracellular proteins. The expression level of MMP is low in the normal brain. However, the expression of MMP is elevated in glioma tissue, and a strong relationship between glioma cell invasiveness and the expression level of MMP has been identified [62,63,64]. Some subclasses of MMP and the gelatinase subclass to which MMP2 and MMP9 belong are associated with glioma invasion. MMP2 interacts with integrin and activates glioma invasiveness [65]. MMP9 is also expressed in other types of surrounding cells, such as macrophages and microglia, and helps to infiltrate and enlarge the tumor mass [66]. MMPs are activated by plasmin and cathepsin B [35,67]. MMP9 is also activated by the mitogen-activated protein kinase (MAPK)/ERK signaling pathway, which is controlled by the epidermal growth factor receptor (EGFR) variant type III [62,68]. MMP activity is regulated by tissue inhibitors of metalloproteinases (TIMPs) [69]. Another subclass of MMPs that contributes to glioma cell invasion is MT-MMPs. MT-MMPs are not only surface activators of pro-MMP2 but also show proteolytic activity toward ECM molecules [70,71]. MT-MMPs are membrane-bound, as their name implies, and they can be activated intracellularly and on the cell surface [72,73]. The expression level of MT-MMPs is associated with the grade of brain tumors and a significant increase in GBMs [71]. ADAMs, another unique subfamily of MMPs, are known to contribute to glioma invasion. ADAMs exhibit both adhesion and proteolytic activities, and they can be classified as soluble ADAMs, transmembrane ADAMs, and ADAMs with thrombospondin domain (ADAMTS) [74]. Some types of ADAMs have been localized in the normal adult human brain [75] and are activated in glioma. Membrane-anchored ADAM12 is strongly expressed in GBM and sheds heparin-binding epidermal growth factor (EGF) [76]. EGF contributes to GBM proliferation through the EGFR signaling pathway [77]. Another study has reported that ADAM12 contributes to glioma invasion by activating MMPs [78]. Reverse transcription-polymerase chain reaction analysis indicated an elevated expression of ADAMTS4 and 5 in GBM [79]. ADAMTS4 and 5 promote growth factors and cytokines and upregulate the invasive potential of glioma cells [79,80,81]. Although several candidate agents that inhibit MMPs have been identified as therapeutic drugs for glioma, none has been used in clinical practice [82,83].

Other enzymes involved in glioma invasion are cysteine proteinases and serine proteases, such as cathepsin B and uPA. Cathepsin D also contributes to the activation of cathepsin B [35]. Previous studies have revealed that cathepsin B contributes to glioma invasion by activating uPA and MMPs and suppressing TIMPs [84,85]. The expression level of cathepsin B correlates with the malignancy of glioma tissues [86,87]. Previous research has revealed that both uPA and uPA receptors (uPAR) activate cell invasion in different ways. uPA activates plasminogen, as its name implies, and releases plasmin, which indicates proteolytic activity and activates MMPs [35,88]. uPAR is expressed in almost all cells and is anchored to the cell membrane. uPAR contributes to intracellular and extracellular pathways, such as the MAPK/ERK and JAK/STAT pathways [89,90,91,92], which contribute to the proliferation and invasion of glioma. Previous studies have also reported that the activation of uPA is increased in malignant glioma and that the expression level of uPA is associated with prognosis in patients with glioma [93,94].

## 3. Interaction with Surrounding

### 3.1. Astrocyte and Oligodendrocyte

The microenvironment surrounding GBM is composed of noncancerous cells, including astrocytes, oligodendrocytes, biomolecules, and ECM, which are crucial for tumor progression. Both parenchymal and nonmalignant cells adjacent to the mass promote GBM development and growth [95]. Astrocytes and oligodendrocytes constitute the majority of the glial cells in the adult brain [96]. Astrocytes have been reported to be involved in glioma progression by interacting with glioma cells. In contrast, although oligodendrocytes and oligodendrocyte progenitor cells have been suggested as possible sources of GBM [97,98], their function in supporting GBM cells has not been thoroughly investigated [99]. Therefore, in this section, we discuss the role of astrocytes in glioma invasion.

#### 3.1.1. Role of Tumor-Associated Astrocytes (TAAs)

Astrocytes are involved in the formation and processing of brain circuits through various cellular functions, including synapse formation, maturation and removal, ion homeostasis, neurotransmitter clearance, regulation of extracellular capacitance, and regulation of synaptic activity [96]. Furthermore, these specialized glial cells contribute to the formation of the blood–brain barrier (BBB) [100,101]. The gene signatures of different astrocyte subpopulations in specific brain regions correlate with glioma tumors containing different genomic alterations. This finding suggests that these cells play specific roles in interacting with the surrounding GBM microenvironment [102,103]. In other words, the heterogeneity of endogenous astrocytes and possibly the local brain microenvironment contributes significantly to the development of gliomas [104]. Astrocytes exhibit a reactive phenotype upon contact with tumor cells and transform into reactive astrocytes through a process called astrogliosis [105]. Reactive astrocytes are characterized by cell hypertrophy (they become hypertrophic); increased expression of intermediate filaments consisting of nestin, vimentin, and glial fibrillary acidic protein (GFAP); and activation of cell proliferation [106]. TAAs, reactive astrocytes in direct contact with GBM cells, increase the malignancy of GBM by facilitating tumor migration, invasion, and proliferation [107,108,109] (Figure 3).

#### 3.1.2. Activation of the Signaling Pathway Associated with Invasion by TAAs

TAAs promote glioma invasion through the activation of several signaling pathways, such as nuclear factor kappa-B (NF-κB), interleukin (IL)-6/JAK/STAT, and sonic hedgehog (SHH) signaling (Figure 3). NF-κB induces one of the major signaling pathways involved in glioma–astrocyte crosstalk, which promotes tumor invasion. First, NF-κB may be involved in the activation of astrocytes in response to TAA during GBM development. Receptor activator of NF-κB ligand (RANKL), a member of the tumor necrosis factor family of cytokines, acts as a ligand for the receptor activator of NF-κB, and the number of activated astrocytes is markedly increased in the peripheral lesions of invasive tumors where RANKL is highly expressed. These astrocytes have been reported to secrete various factors that regulate glioma cell invasion, including transforming growth factor-β (TGF-β) [110]. Additionally, connexin 43 (Cx43), the principal astrocytic gap junction protein, promotes the formation of GBM-TAA gap junctions. This protein activates the stimulator of the interferon gene pathway through the transfer of the second messenger 2′,3′-cyclic guanosine monophosphate–adenosine monophosphate (cGAMP). The transfection of cGAMP from GBM cells to TAA results in the production of factors such as tumor necrosis factor (TNF) and interferon-α by astrocytes. These factors activate NF-κB and STAT1 in GBM cells, ultimately supporting tumor cell growth and invasion [111,112,113]. Under hypoxic conditions, astrocytes secrete chemokine C–C motif ligand (CCL) 20, which binds to chemokine C–C motif receptor 6, stimulates the NF-κB signaling pathway, and enhances the expression of HIF-1α in GBM cells, thereby promoting GBM cell invasion and proliferation [114].

Neuron-derived SHH signaling is mediated by the Gli family of zinc finger transcription factors that regulate the expression of target genes and ultimately result in the activation of TAAs [115]. Suppression of SHH-Gli signaling markedly inhibits glioma cell migration and invasion [116], and SHH-Gli signaling may promote astrocyte activation in the perivascular niche surrounding the glioma, thus facilitating glioma invasion.

The IL-6/JAK/STAT signaling pathway is one of the main signaling pathways involved in tumor migration and invasion. The co-culture of astrocytes and glioma cells enhanced IL-6 secretion by TAAs, thus promoting GBM migration and invasion [117]. Furthermore, IL-6 activates STAT3 [118], which promotes GBM invasion and angiogenesis through the JAK/STAT3 pathway [119]. Recent reports have indicated that a mutual activation loop is formed between astrocytes and GBM cells via IL-6/STAT3 signaling, which promotes GBM cell invasion, proliferation, and resistance to apoptosis [12,120].

Furthermore, TAAs secrete glial cell line-derived neurotrophic factor (GDNF), which activates receptor tyrosine kinase/glial cell line-derived neurotrophic factor family receptor 1 expressed by GBM cells, thereby activating the downstream PI3K/Akt pathway to promote GBM cell invasion [121]. A recent report suggested that the induction of tunneling nanotube structures between GBM cells and surrounding non-neoplastic astrocytes may ultimately lead to the adaptation of non-neoplastic astrocytes to tumor-like metabolism and hypoxic conditions, thus increasing tumor invasiveness and contributing to tumorigenesis [122].

#### 3.1.3. The Crosstalk between Glioma Cells and Astrocytes via Extracellular Vesicles (EVs) Contributes to Glioma Invasion

Another mechanism by which TAAs promote glioma cell invasion is the secretion of EVs. EVs have been reported to have multifaceted functions in neuronal and glial cell crosstalk and play an important role in communication between cell types within GBM and its microenvironment [123] (Figure 3). EVs are nanospheres containing proteins, lipids, and nucleic acids, such as deoxyribonucleic acid (DNA), messenger ribonucleic acid (mRNA), and non-coding RNA, which are released from most cells [124]. EVs play an important role in cell-to-cell communication by allowing access and delivery to adjacent tumor cells and distant sites [125]. It has been reported that GBM-derived EVs are also taken up by astrocytes, and EVs derived from surgically resected primary GBM tumors transform primary human astrocytes into reactive astrocytes [126]. Astrocytes exposed to GBM-derived EVs secrete and overexpress MMPs such as MMP2, MMP9, and MMP14 [127]. In addition, there is increasing evidence that astrocytes also use EVs for signaling and affect GBM, such as the report that EVs shed from astrocytes reduce the level of the tumor suppressor gene phosphatase and tensin homolog deleted from chromosome 10 in GBM tumor cells [128].

Therefore, it is important to elucidate the mechanism of crosstalk between astrocytes, oligodendrocytes, and glioma cells. TAAs have been found to play multiple roles in invasion, tumor survival, and proliferation, and further research into the mechanisms involved in TAA–GBM crosstalk is expected to yield novel markers and targets for future GBM therapy.

### 3.2. Microglia

#### 3.2.1. Introduction of Glioma-Associated Macrophages/Microglia (GAM)

Macrophages and microglia have been the focus of much attention in the glioma microenvironment. In GBM, GAM represents the major population, comprising up to 50% of the cells of the tumor mass. Bone marrow-derived macrophages and monocyte cells (BDMC) account for 85% of the total GAMs, while resident microglia account for the remaining 15% [129]. Interestingly, microglia-derived GAMs predominate in newly diagnosed gliomas. However, after relapse, they are outnumbered by monocyte-derived GAMs, especially in hypoxic environments [130]. The total number of microglial cells does not change with the grade of malignancy. However, macrophage-like cells gain ascendancy in high-grade gliomas. Moreover, microglia correlate with poorer survival in GBM when considering clusters of differentiation (CD) 163^+^ cells, a pro-tumoral macrophage that promotes the development of tumors, whereas it does not change prognosis in isocitrate dehydrogenase-mutated low-grade gliomas [131]. Initially, upon activation, GAMs were classified into two phenotypes: a pro-inflammatory M1 phenotype, and an anti-inflammatory M2 phenotype. Historically, in the context of GBM, GAMs have been considered to possess an M2-like phenotype [132]. However, with a better understanding of the population, it is highly likely that GAMs are composed of heterogeneous subpopulations [133,134]. In the tumor core, GAMs evolve into a pro-inflammatory state, and a subpopulation of cells is identified with a strong, opposing correlation to programmed cell death-1 (PD-1) signaling, which may correlate with their response to PD-1 inhibition. The co-stimulatory molecule CD80/CD86, expressed by GAMs, interacts with cytotoxic T-lymphocyte-associated antigen 4, causing reduced T-cell activation [135]. Additionally, in the peritumoral area, GAMs evolve toward anti-inflammatory phenotypes and contain a population of cells strongly associated with NF-kB signaling. These results advocate the need for a multi-targeted approach for the treatment of GBM [136].

#### 3.2.2. Promotion of Tumor Invasion by GAMs

GAMs have been shown to release a number of factors that stimulate tumor growth and invasion, including TGF-β, IL-1β, IL-6, stress-inducible protein 1 (STI1), and EGF, by acting on GICs in the perivascular niche and glioma cells (Figure 4).

In the tumor microenvironment, GAMs release factors that lead to degradation of the extracellular matrix and stimulate signaling pathways to promote glioma cell invasion. GICs express higher levels of type II TGF-β receptor mRNA and protein. TGF-β is secreted from GAMs and causes the production of MMP9, which disrupts the ECM and increases the invasion of GICs [137]. Proneural GBM patients with high IL-1β expression showed shorter survival than patients with low IL-1β levels. Overexpression of IL-1β in GAMs results in the activation of the p38 MAPK pathway and production of CCL2 (or MCP-1), which promotes GICs proliferation [138]. In addition, the GAM-induced release of STI1 and EGF promoted GBM growth and invasion [139,140]. M2-polarized microglia but not BDMCs, induced platelet-derived growth factor (PDGF) receptor beta expression in glioma cells, and stimulated their migratory capacity and, consequently, tumor progression [141]. Loss of osteopontin in GAMs but not in glioma cells enhances tumor progression [142]. TGF-β2, expressed by GAMs, induces MMP2 expression and blocks TIMP-2, promoting tumor invasion [143]. CCL8 (or MCP-2) was found to be highly expressed in GAMs and was correlated with poor survival. In a murine GBM model, CCL8 was shown to promote invasion and increase tumor cell stemness via the ERK1/2 signaling pathway [144]. As innate immune cells, GAMs express Toll-like receptors (TLRs), which are pathogen recognition receptors that play an active role in tumor growth. For instance, the expression of TLR2 is increased in GAMs, and its ablation results in better prognosis [145]. TLR2 promotes GAM production of MMP14, which is essential for MMP2 release and glioma invasion, and microglial expression of MT-MMP, which cleaves proMMP2 into its active form, leading to tumor invasion [146,147].

#### 3.2.3. Roles of GAMs for Changing Microenvironments

Using nanostring analysis, the subtypes of human-established GBM were examined for the expression of allograft inflammatory factor 1 (AIF1; encoding IBA1) as a marker for GAMs. It was found that AIF1 expression is higher in the mesenchymal subtype, which is the most aggressive and strongly associated with a poor prognosis, than in the proneural and classical subtypes. Furthermore, AIF1 expression was found to affect survival depending on the subtype of GBM analyzed. Higher AIF1 levels were associated with longer survival in the mesenchymal subtype, whereas higher AIF1 expression was associated with shorter survival in proneural tumors [148]. Moreover, it has been reported that PMT is associated with increased GAMs and activation of the TNF/NF-κb signaling pathway [149]. Another mechanism by which GAMs contribute to PMT in GICs is by secreting EVs, which are also secreted from TAAs and contribute to changes in the microenvironments of glioma through their contents, including mRNA [123,124]. EVs from monocyte-derived macrophages transfer microRNA (miR)-27a-3p, miR-22-3p, and miR-221-3p to GICs. These miRNAs promote several mesenchymal phenotypes in proneural GICs by simultaneously targeting chromodomain helicase DNA-binding protein 7 (CHD7), which regulates neural stem cell maintenance and development [150].

Tumor cells release chemoattractant proteins that recruit GAMs, such as CCL2, colony stimulating factor-1 (CSF-1), granulocyte–macrophage colony stimulating factor, hepatocyte growth factor or scatter factor, stroma-derived factor 1 (SDF-1), and GDNF (Figure 4). GBM cells express high levels of CCL2, which recruits GAMs and promotes tumor growth [129]. However, other studies have shown that CCL7 (or monocyte chemoattractant protein (MCP) -3), but not CCL2, acts as a GAM chemoattractant [151]. Glioma-derived IL-33, in its secreted form, has been found to be associated with the recruitment and invasion of GAMs [152]. In addition, CSF-1 (alternatively called macrophage-CSF) has been shown to promote GAM motility and induce the switch to a more immunosuppressive phenotype.

GAMs control tumor angiogenesis by sensing hypoxic conditions, producing IL-1β, and increasing VEGF-A expression, which are regulators of vascular permeability and tumor angiogenesis [153]. Glioma cells also control angiogenesis through EVs. Wilms’ tumor 1 in glioma-derived EVs serves as a potent promoter of tumor progression by inhibiting microglial expression of the Thbs1 gene and acts as an anti-angiogenic factor, which may lead to enhanced angiogenesis in glioma [154].

GAMs are likely to contribute to tumor survival against radiation therapies and chemotherapies by initiating regenerative programs such as wound healing and tumor recurrence. The HIF-1 signaling pathway has been proposed to be involved in resistance to radiation therapy. In GAMs, radiation-induced expression of HIF-1 leads to the activation of SDF-1 and its cognate receptor C–X–C motif chemokine receptor type 4 [155], causing monocyte recruitment, angiogenesis, and tumor recurrence [156]. Radiation therapy has the capacity to increase the recruitment of monocytes via CSF-1 [157].

### 3.3. Glioma Cell

The interaction between glioma cells and GICs is one of the factors that comprise the tumor microenvironment is the interaction between glioma cells and GICs. This interaction regulates malignant phenotypes such as cell proliferation, migration, invasion, stemness maintenance, and therapeutic resistance. The cell-to-cell interaction systems known in glioma are the tumor cell membrane tube network, erythropoietin-producing human hepatocellular receptor (Ephs)–ephrin signaling, and Notch signaling (Figure 5).

#### 3.3.1. Tumor Cell Membrane Tube Network

Astrocytic tumor cells extend ultralong membrane protrusions. In glioma, cell-to-cell connections by membrane tubes form a multicellular network, which makes it possible to interconnect distant glioma cells. This tumor membrane tube network is associated with tumor invasion, cell proliferation, self-repair systems, and resistance to radiotherapy. Cx43, a gap junction protein in GICs, plays a key role in this system. The suppression of Cx43 in the periphery of glioma cells can reduce the invasive capacity of glioma cells and may be beneficial for distal tumor recurrence. However, restoring Cx43 in GICs can inhibit GICs self-renewal and tumor initiation [158,159]. Therefore, targeting the tumor membrane tube network regulated by Cx43 may be a new approach against glioma progression. Recently, it was shown that Cx43 expression levels in proneural subtype GICs were higher than those in mesenchymal subtype GICs [159]. Upregulated Cx43 modulates E-cadherin, which is an important protein in PMT and suppresses stem-related cell surface markers such as sex determining region Y box 2, c-Met, and CD133 [160,161,162]. Cx43 upregulation may prevent the malignant phenotype of GICs with Cx43 low expression [163].

#### 3.3.2. Ephs–Ephrins Pathway

Eph and ephrins play primary roles in embryogenesis and early development (Figure 5). The Eph–ephrin system belongs to the largest receptor tyrosine kinase family, which is activated by cell-to-cell contact. Signal transduction through the Eph–ephrin system is generated bi-directionally upon ligand–receptor binding, initiating ‘‘forward signaling’’ through Eph phosphorylation and ‘‘reverse signaling’’ through ephrin activation. The intensity of this pathway, generated in response to receptor activation, greatly depends on the nature of ligand stimulation. Ephs have been divided into two groups: EphA (A1–A8, A10) and EphB (B1–4, B6) [164,165,166]. The Eph–ephrin system plays an important role in malignant phenotypes of cancers [166]. Activation of the Ephs–ephrins pathway occurs when the binding cell surface ephrin induces phosphorylation of the Eph intracellular domain, resulting in the activation or inhibition of downstream signals such as the MAPK/ERK and PI3K/Akt pathways. Previous studies have described various functions of the Ephs–ephrins system in glioma [165,166]. EphA family receptors are expressed in GICs but are absent in less aggressive differentiated glioma cells. Elevation of either EphA2 or EphA3 maintains glioma cells in a stem-like state by negatively regulating the MAPK/ERK pathway. EphA2 and EphA3 were highly expressed in the mesenchymal subtype glioma according to The Cancer Genome Atlas database [164]. The EphA family of receptors may be associated with a switch to a more aggressive phenotype, the mesenchymal phenotype. In particular, EphA2 is not expressed in normal brain tissue but is highly expressed in more than 90% of gliomas. Therefore, EphA2 may be a specific tumor marker of glioma and is considered an important factor in glioma development [167,168]. The high-affinity EphA2 ligand, ephrin-A1, is expressed at a low level in EphA2 positive areas in glioma [167,169]. Further research has revealed that EphA2 and ephrin-A1 are also expressed in other solid tumors and contribute to tumorigenesis, angiogenesis, and tumor invasion [170]. Overexpression of ephrin-A1 downregulates EphA2 and FAK, leading to reduced migration, adhesion, and proliferation of glioma cells [167]. Elevated ephrin-A1 expression may lead to a less aggressive phenotype in differentiated glioma cells. EphB/ephrin-B signaling is related to the adhesion, migration, and invasion of glioma cells [171]. High expression of EphB2 can reduce adhesion and increase the migration and invasion of glioma cells [171]. Gliomas derived from GIC overexpressing EphB2 have a high invasive potential. The EphB ligands ephrin-B2 and B3 enhance cell migration and invasion [172,173]. High ephrin-B2 expression is a strong predictor of a poor prognosis [173]. EphB4/ephrin-B2 is highly expressed in many malignant tumor cells, including glioma cells. EphB4 can promote tumor angiogenesis through Notch signaling [174,175]. Recently, it was shown that EphB4 receptor activation by ephrin-B2 suppresses glioma migration and invasion via the suppression of Akt phosphorylation [176]. EphB4/ephrin-B2 signaling may occur inefficiently in peripheral tumor sites of low cellularity, where cell–cell contact is rare. Therefore, EphB4 signaling retains glioma cells in the central portion of the tumor via a positive feedback loop, and liberation of the cells from EphB4 signaling may be responsible for their invasion into the brain [176]. However, almost all subtypes of the Ephs–ephrins pathway are involved in glioma progression and are being investigated as a therapeutic target [167,177]. EphB1 signaling suppresses glioma invasion and is associated with a favorable prognosis [178]. Recently, preclinical trials have been conducted targeting EphA2, which is highly expressed in gliomas, and anti-tumor effects have been observed through EphA2 suppression [168].

#### 3.3.3. Notch Signal

Notch signaling plays a variety of roles in cell differentiation and the maintenance of stemness in the central nervous system [179]. However, this signal could be related to cancer initiation, propagation, and invasion and is critical for the maintenance of stemness in various cancers, including glioma [180,181,182]. This pathway is mediated by cell-to-cell contact, thereby initiating the binding of a ligand to a Notch receptor with subsequent activation of intracellular signaling events. Upon ligand binding, the Notch intracellular domain released by γ-secretase translocates to the nucleus, interacts with specific transcriptional factors, and subsequently activates downstream signals [181,182] (Figure 5). Notch signaling displays crosstalk with other oncogenic pathways such as the PI3K/Akt, JAK/STAT, SHH, and Wnt pathways. Therefore, Notch inhibition has been shown to be effective in the initiation and progression of gliomas [180,183,184,185,186]. Notch signaling has been shown to regulate phenotypic changes in GICs through the IL6/STAT3 pathway [12]. Hence, the suppression of Notch signaling may offer an ideal strategy for the treatment of glioma, and clinical trials of Notch inhibitors are being developed for patients with recurrent, progressive gliomas [187,188].

In gliomas, these cell-to-cell systems may be intricately interrelated within the tumor. Gliomas may create a convenient tumor environment using these cell-to-cell systems and develop tumor progression and therapeutic resistance. Therefore, regulation of cell-to-cell systems may be key to the development of novel glioma therapies.

## 4. Alteration of the Microenvironment Affected by Therapy

The standard of care for patients with GBM is surgical resection followed by chemotherapy and radiotherapy. Herein, we discuss the mechanisms that underlie the alteration of the tumor microenvironment induced by surgery, chemotherapy, and radiotherapy. Interestingly, some of these mechanisms overlap and contribute to glioma invasion in complementary ways. There are two main pathways for this alteration: changes in the surrounding environment and the phenotype of glioma cells. All of these treatments against glioma have both advantages and disadvantages (Figure 6).

### 4.1. Operation

Surgical resection is essential in glioma therapy. Previous studies have described the clinical advancement of surgical resection of gliomas [189,190,191]. However, some reports revealed unfavorable pathological findings in residual tumors, such as increased proliferation and prominent invasion [192,193]. The mechanism underlying the increased aggressiveness of recurrent glioma remains unclear. Only a few basic studies have focused on the alteration of the microenvironment of the residual glioma because of the difficulty in establishing appropriate preclinical models [193,194]. These previous studies suggest that reactive astrocytes, microglia, and macrophages, which are stimulated by surgical incision, promote the proliferation and migration of glioma [95,194,195] (Figure 6). Some basic research has discussed the molecular mechanisms underlying the alteration induced by reactive astrocytes [112,196]. Reactive astrocytes promote glioma invasion in a paracrine manner via Cx43 [112,197]. RNA sequencing and gene expression analysis in injured astrocytes revealed that the expression of C–X–C motif chemokine ligand (CXCL) 5 contributes to glioma invasion [194]. Another study revealed that the postoperative activation of CXCL5 signaling and miR451/liver kinase B1/AMP activated protein kinase/organic cation transporter 1/mammalian target of rapamycin pathway enhanced glioma cell invasion. However, the interaction between these two signaling pathways remains unclear [196]. Other studies have reported the contribution of microglia and macrophages to glioma migration [95,195]. Because of the low proliferative activity and inherent resistance to cytotoxic therapies, the establishment of appropriate treatments for these cells that contribute to glioma invasion is difficult [95]. The association between glioma recurrence and angiogenesis induced by tumor resection remains unclear. Studies with preclinical glioma resection models revealed that the expression level of Ki-67, the marker for cell proliferation, is statistically higher in the specimens of recurrent gliomas; on the other hand, CD31, the initiative marker for blood vessels, is statistically lower [193].

### 4.2. Chemotherapy

#### 4.2.1. Temozolomide (TMZ)

TMZ, an alkylating agent and dacarbazine, is the first-line treatment drug for patients with GBM. The anti-cancer effects of TMZ against glioma provoke the failure of DNA synthesis by TMZ-induced O-6-methylguanine adducts and the inhibition of O-6-methylguanine-DNAtransmethylferase (MGMT) [198]. TMZ treatment could improve the prognosis of GBM patients [2,199]. However, previous studies have revealed that long-term TMZ treatment induces changes in glioma cells and chemoresistance [198]. Changes in glioma cells are associated with their microenvironment and invasiveness. In basic research, TMZ-treated human glioma cells demonstrated more migration or invasion activities than untreated cells [200]. The most important changes caused by TMZ treatment are the conversion to cancer stem cells and provocation of PMT (Figure 6). TMZ-induced GICs express CD133 and CD44, which are known to initiate stemness markers [201]. Interestingly, some previous papers have mentioned the contribution of GICs to resistance to chemotherapy and tumor recurrence. However, these GICs were not associated with tumor invasion [200,202]. A further study reported that the TMZ resistance glioma cells, which strongly express multidrug resistance protein 1 (MDR1), enhanced cell migration [200]. MDR1 is associated with PMT and endothelial mesenchymal transition in some cancers [175,203,204]. Therefore, these findings suggest that TMZ treatment induces the expression of MDR1, and MDR1 contributes to PMT and aggressive invasiveness of glioma. Previous studies have reported that the expression of Cx43 in glioma cells after TMZ treatment is also associated with glioma invasion [205,206]. Cx43 inhibits TMZ cytotoxicity and glioma cell death via the mitochondrial apoptosis pathway [201,205,207]. Another mechanism of glioma invasion involves damage to the brain ECM induced by TMZ treatment. TMZ disorganizes the construction of proteoglycans, induces fragile extracellular structures, and contributes to glioma invasion [208,209].

#### 4.2.2. Bevacizumab (BEV)

BEV, a VEGF inhibitor, is also a known therapeutic agent for GBM. Blockage of VEGF disrupts the perivascular niche and inhibits tumor growth [95]. Some clinical trials have revealed that BEV treatment reduced tumor size and prolonged progression-free survival, but no effect on overall patient survival was observed [210,211,212,213]. Some studies have revealed that the microenvironment surrounding glioma cells changes and activates invasiveness following BEV treatment. There are three main mechanisms for the activation of invasiveness in BEV-treated glioma cells: the provocation of acidity in the tumor tissue, induction of glioma initiating myeloid cells, and changes in the phenotype of glioma cells [214] (Figure 6). In GBM tissue, the VEGF family pathway is activated for angiogenesis, and BEV treatment induces hypoxia in tumor tissue by reducing tumor perfusion [215]. Under hypoxic conditions, glioma cells reduce glucose metabolism, which is associated with the tricarboxylic acid cycle and improves glycolysis and the PPP. The upregulation of glycolysis causes an increase in lactate levels in tumor tissue and induces microenvironmental acidosis [214]. Acidity and low pH are known to be drivers of glioma invasion [216,217]. Previous research has revealed that tumor cell invasion increases in low pH level models [216]. Another mechanism of growth advantage under hypoxic conditions is the induction of glioma initiating myeloid cells [218,219]. Previous studies have reported that glioma initiating myeloid cells such as macrophages and microglia contribute to angiogenesis and invasion [218]. These macrophages are known as tumor-supportive macrophages (M2) and express STAT3 [220]. According to these findings, some basic research has mentioned additional therapies targeting the evasion mechanism of BEV treatment. The phosphorylation of STAT3, especially at the tyrosin-705 site, was observed in the BEV-treated glioma specimens, and the additional blockage of STAT3 significantly reduced the tumor size [219]. Other studies have also revealed that anti-angiogenic therapy treatment with BEV induced the two subtypes of GBM [221]. One glioma cell subtype, which showed enhanced proliferation and unchanged invasiveness, expressed mitogen-activated protein kinases, neural cell adhesion molecule 1, and aquaporin 4. The other subtype of glioma cells, which showed unchanged proliferation and enhanced invasiveness, expressed HIF-1α, fibronectin, CXCL12, and PDGF-β [221]. The latter subtype of glioma cells is located at hypoxic lesions and is specialized for invasion. In contrast to TMZ treatment, the expression of stem cell markers, CD133 and CD44, remained unchanged under BEV treatment; thus, glioma cell profiles associated with stemness would remain stable [214].

### 4.3. Radiation

Radiotherapy improves overall patient survival by controlling patients’ symptoms, tumor recurrence, and progressing BBB permeability [222,223,224,225]. However, radiotherapy remodels the glioma microenvironment. Previous research has revealed two main mechanisms by which radiation therapy contributes to the invasion of glioma cells: damage to normal brain networks, and transition of glioma cell phenotypes. Six months or more after radiation therapy, delayed irreversible and progressive brain injury occurs [226,227]. The mechanism of brain injury is damage to the normal brain parenchyma, including oligodendrocytes, neuron stem cells, and vascular endothelial cells [226,227]. Delayed brain injury induces demyelination, neuroinflammation, and chronic ischemia and leads to the loss of normal neuron–glia networks [228]. An altered brain environment with fragile brain networks permits the aggressive invasion of glioma cells [229]. Another mechanism of glioma invasion induced by radiation therapy is associated with PMT. A previous study revealed that the specimens of recurrent glioma after radiation therapy activated the expression levels of mesenchymal markers, such as collagen 1A, alpha-smooth muscle actin, CD44, and YKL-40 and reduced the expression of GFAP, a glial marker [230]. Some basic research has also mentioned that glioma spheres exposed to radiation tend to activate invasiveness. This trend was associated with the activation of STAT3, which is a marker of the mesenchymal phenotype [231,232]. These findings support the hypothesis that the activated invasiveness of irradiated gliomas is associated with PMT. Radiation therapy induces changes in the microenvironment within the tumor and border of normal brain tissue. Mesenchymal phenotype cells are typically observed in the tumor-invasive region, and this finding suggests that PMT is associated with the tumor microenvironment, such as hypoxia and neuroinflammation [26,38,40,233]. Further research revealed that there are two main signaling pathways that contribute to glioma PMT after radiation therapy. The surrounding macrophages or microglia release growth factors, such as TGF-β, EGF, PDGF, and fibroblast growth factor-2 [230], and these molecules evoke the Smad1/2 pathway. ERK1/2 signaling, which is enhanced by reactive oxygen, is another pathway [233]. These molecules accelerate the PMT of gliomas via the Snail pathway [230,233].

## 5. Perspective

Some clinical trials focusing on the mechanism of glioma invasion have been conducted [83,234]. Marimastat, a popular mimetic inhibitor of the MMP family of enzymes, has been used in clinical trials as a therapeutic drug for GBM. This was a multicenter, randomized, double-blind, placebo-controlled trial. The results revealed that there was no clinical benefit for survival between the two groups, and 20% of patients who underwent marimastat treatment experienced musculoskeletal toxicity as a side effect [83]. Cilengitide, an inhibitor of the αvβ3 and αvβ5 integrin receptors, has been used in clinical trials [234,235]. A phase II clinical trial in patients with recurrent GBM revealed anti-cancer effects without severe side effects [235]. Based on this result, a multicenter, randomized, open-label, phase III clinical trial that conducted cilengitide on primary GBM (CENTRIC EORTC 26071-22072 study) was performed [234]. Patients newly diagnosed with GBM with a methylated MGMT promoter were randomized, and the cilengitide treatment group had an integrin inhibitor added to their chemoradiotherapy regimen. Despite the results of a phase II clinical trial, there was no clinical benefit identified with the addition of cilengitide in a phase III study [234]. A phase I clinical trial of combination therapy with cilengitide and cediranib, an anti-angiogenic drug, was then performed. However, no survival or response benefits were observed [236].

According to the results of these studies, some problems regarding anti-invasive drugs for clinical use have been debated. One of the major problems is the penetration of the candidate drugs into the BBB. The BBB consists of a multilayered barrier that selectively permeates materials from the blood to the brain in normal brain tissue [237]. However, malignant brain tumors damage normal brain structures, including the BBB, and some previous reports indicate that their invasive nature would work as an advantage for us because the drug concentration in glioma tissue is similar to the blood concentration [238]. Previous studies have mentioned that cilengitide could indicate the effect of normalization of the BBB [234]. Therefore, anti-invasion treatments would suppress the progression of glioma cells and decrease the penetration of therapeutic drugs. Another problem is the insufficient review of images. Each clinical trial focused on the prognosis and side effects. However, no detailed evaluations of images, such as the secular change around the edge of glioma, have been mentioned [83,234]. Although no drugs have been successfully applied clinically, some candidates have been identified. Chotemin, a specific HIF inhibitor, revealed anti-cancer effects for glioma cells and GICs in basic research. Further investigations are warranted to identify new candidate drugs for patients with gliomas [239,240,241].

Other clinical treatment aids for glioma, which focus on the microenvironment, have been explored. Several previous studies on viral therapy that exerted anti-tumor effects on glioma by oncolytic viruses have indicated some changes in the microenvironment. Viral therapies induce increased permeability of the vasculature surrounding the glioma and enhance inflammation through cytokine reactions [242]. The tumor treating field (TTF), which is derived from electric fields via non-invasive transducer arrays, alters the microenvironment surrounding gliomas [243,244]. TTF-treated cells, which induce aberrant mitosis, release cellular stress signals such as the endoplasmic reticulum chaperonin calreticulin. These cellular stress signals enhance immune reactions [244]. However, the alteration of the microenvironment by TTF is still unknown; thus, further analysis is required.

As mentioned above, it appears that normal brain reactions to gliomas tend to scatter tumor cells in the normal brain. Tumor microenvironments promote the transition of GBM cells from mesenchymal to proneural, which is more invasive. The former subtype of GBM has a poorer prognosis than the latter [26,38,40,245]. According to the results of anti-invasion clinical trials, we should reconsider whether anti-invasion treatments for glioma improve prognosis. Anti-invasion treatments may maintain the original brain-specific defense mechanism and protect glioma cells from anti-cancer candidates [234]. Therefore, the contribution of glioma invasiveness to prognosis is debatable. It is necessary to reconsider whether the invasion itself should be considered beneficial or not. Further studies in this area are warranted to confirm this paradox and improve the treatment provided to patients with glioma.

## Figures and Tables

**Figure 1 brainsci-12-00505-f001:**
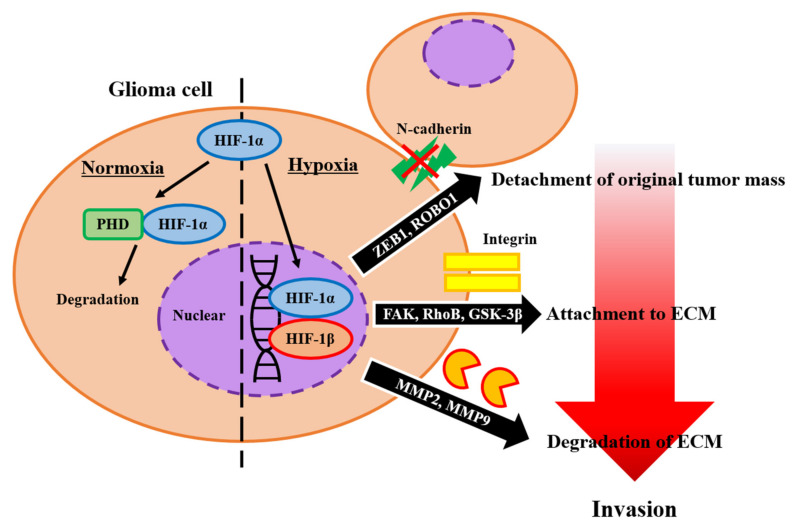
Schematic presentation of hypoxia-inducible factor-1 (HIF-1) function. In normoxia, prolyl hydroxylase (PHD) induced the degradation of HIF-1. In hypoxic conditions, HIF-1 contributes to several steps of glioma invasion. At first, HIF-1 detaches glioma cells from the original tumor mass by activating zinc finger E-box binding homeobox 1 (ZEB1) and roundabout guidance receptor 1 (ROBO1). In the next step, HIF-1 contributes to attachment to the surrounding extracellular matrix (ECM) by focal adhesion kinase (FAK), Ras homolog family member B (RhoB), and glycogen synthase kinase 3β (GSK-3β). Finally, HIF-1 activates matrix metalloproteinases (MMPs) and degrades ECMs.

**Figure 2 brainsci-12-00505-f002:**
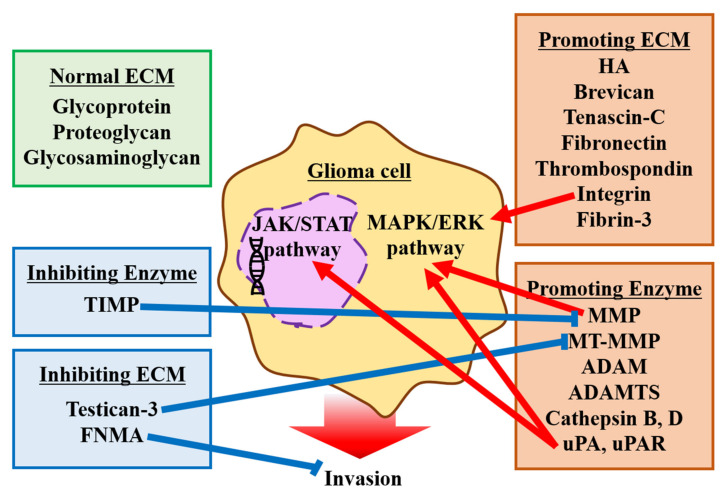
Schematic of the factors affecting the alteration of the glioma microenvironment by extracellular matrices (ECMs) and enzymes. Basically, there are some types of ECM that are independent of glioma cell invasions. However, the alteration of the microenvironment surrounding gliomas is affected by ECMs and enzymes, which almost promote the invasiveness of glioma cells. There are mainly two signaling pathways subserving glioma cell invasion: the Janus kinase (JAK)/signal transducer and activator of transcription (STAT) pathway, and the mitogen-activated protein kinase (MAPK)/extracellular signal-regulated kinase (ERK) pathway. Red arrows indicate the activation of signaling, while blue arrows indicate the inhibition of signaling. Abbreviation: ADAM; a disintegrin and metalloproteinase, ADAMTS; ADAM with thrombospondin domain, FNMA; fibronectin matrix assembly, HA; hyaluronic acid, MMP; matrix metalloproteinase, MT-MMP; membrane-type MMP, TIMP; tissue inhibitor of metalloproteinase, uPA; urokinase-type plasminogen activator, uPAR; uPA receptor.

**Figure 3 brainsci-12-00505-f003:**
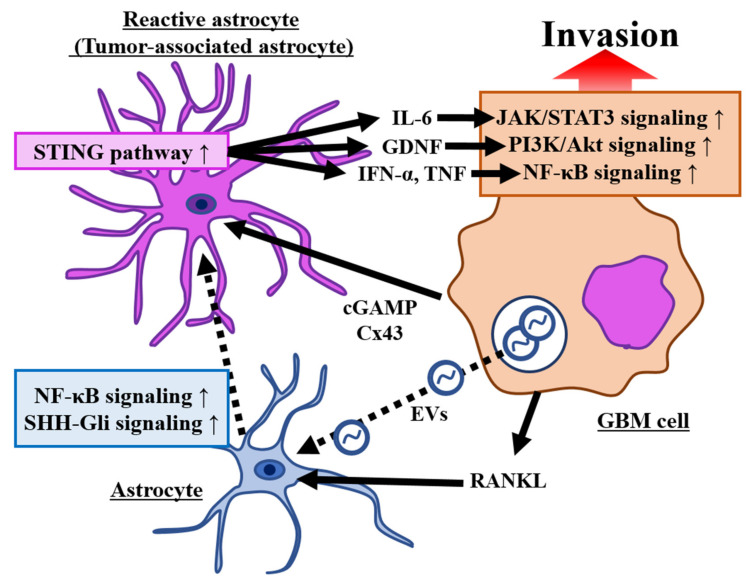
Signaling pathways of glioma–astrocyte crosstalk associated with invasion. The major signaling pathways involved in glioma–astrocyte crosstalk that promote tumor invasion (Black arrows). The activation of nuclear factor kappa-B (NF-κB) and sonic hedgehog (SHH)-Gli signaling pathway induced normal astrocytes into reactive astrocytes (tumor-associated astrocytes (TAAs)). TAAs promote glioma cell invasion by activating some pathways, such as Janus kinase (JAK)/signal transducer and activator of transcription (STAT), phosphatidylinositol-3 kinase (PI3K)/Akt, and NF-κB pathways. The squared arrows show indirect pathways. Abbreviation: cGAMP; 2′3′-cyclic GMP-AMP, Cx43; connexin 43, EVs; extracellular vesicles, GBM; glioblastoma, GDNF; glial cell line-derived neurotrophic factor, IFN-α; interferon-α, IL; interleukin, RANKL; receptor activator of nuclear factor kappa-B ligand, STING; stimulator of interferon genes, TNF; tumor necrosis factor.

**Figure 4 brainsci-12-00505-f004:**
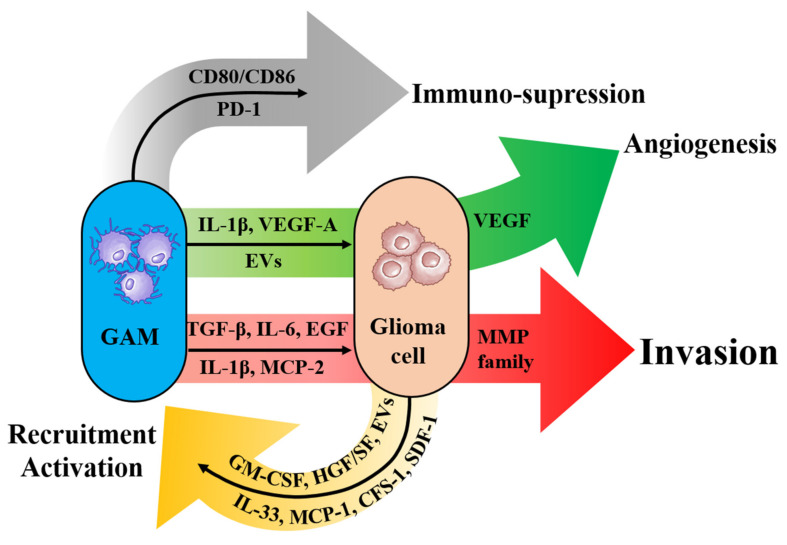
Scheme of the mechanisms of glioma-associated macrophages/microglia (GAM). GAM plays a major role in glioma invasion, angiogenesis, and immunosuppression. The main enzyme contributing to glioma cell invasion is the matrix metalloproteinases (MMP) family. The activation of vascular endothelial growth factor (VEGF) associates with angiogenesis, and programmed cell death-1 (PD-1) is the one of the key molecules for immunosuppression. Glioma cells are involved in the recruitment and activation of GAMs. GAM–glioma cell crosstalk is caused by various chemoattractant molecules and extracellular vesicles (EVs). Abbreviation: CD; clusters of differentiation, CFS-1; colony stimulating factor-1, EGF; epidermal growth factor, GM-CSF; granulocyte-macrophage colony stimulating factor, HGF; hepatocyte growth factor, IL; interleukin, MCP; monocyte chemoattractant protein, MMP; matrix metalloproteinase, SDF-1; stroma-derived factor-1, TGF; transforming growth factor.

**Figure 5 brainsci-12-00505-f005:**
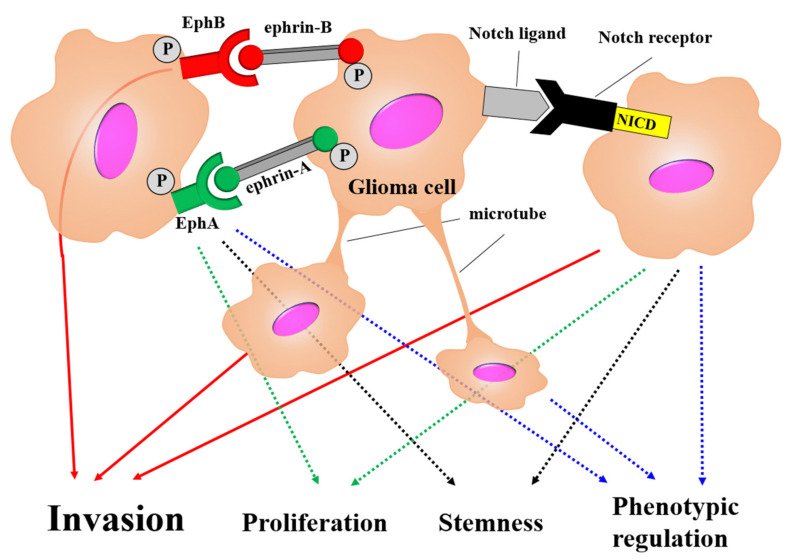
Cell-to-cell systems schema related to the glioma microenvironment. Glioma cells form the tumor environment, developing microtube networks, erythropoietin-producing human hepatocellular receptor (Ephs)–ephrin pathways, and Notch signals. Cell-to-cell systems are mainly associated with invasion (red line), proliferation (green dotted line), stemness maintenance (black dotted line), and phenotypic regulation (blue dotted line) in glioma cells. Invasion ability is acquired from the microtube network, EphB–ephrin-B pathway, and Notch signaling. Cell proliferation and stemness maintenance are involved in the ephrinA–ephrinA pathway and Notch signaling. The glioma phenotype is regulated by the microtube network, EphA–ephrinA pathway, and Notch signaling. P in a grey circle indicate the phosphorylation. Abbreviation: NICD; notch intracellular domain.

**Figure 6 brainsci-12-00505-f006:**
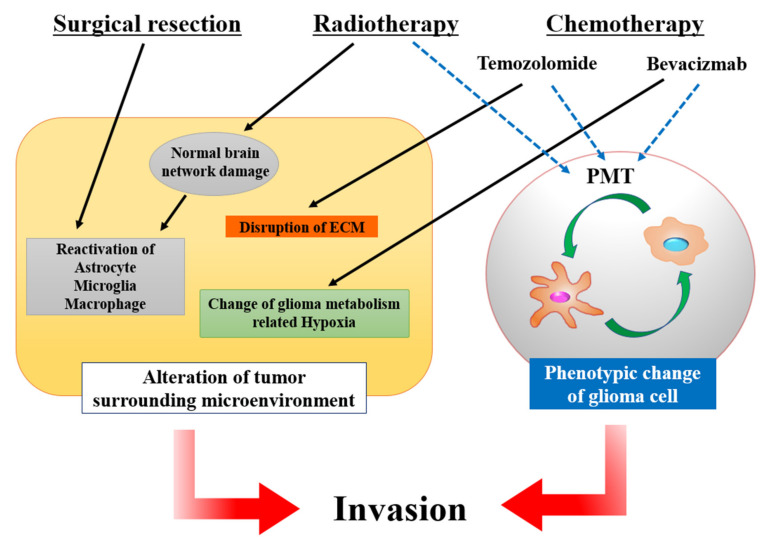
Schema of the mechanisms that underlie the alteration of the glioma microenvironment affected by therapy. Three main therapeutic strategies, namely surgical resection, radiotherapy, and chemotherapy, induce some mechanisms and contribute to glioma invasion in a complementary manner. Some of these mechanisms can divided into two categories: the alteration of the tumor surrounding microenvironment (black arrows), and phenotype change of glioma cells (blue arrows). Proneural mesenchymal transition (PMT) is a main program of phenotype change of glioma cells (green arrows). According to these mechanisms, glioma cells activate invasiveness (red arrows). Abbreviation: ECM, extracellular matrix.

## Data Availability

Not applicable.

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
