# Peer review of "Tumor Microenvironment in Glioma Invasion"

_brainsci, 2022, doi:10.3390/brainsci12040505_

Round 1

Reviewer 1 Report

As the article aims to review current knowledge in glioma, the authors did good work in doing so. The authors have summarized the current knowledge thoroughly and systematically, and the manuscript is well written. This manuscript is suitable for publication, with a couple of text editing.
1. Line 86: should it be N-cadherin instead of ’N cadherin ’?
2. Figure 3, page 8: ‘INF-a’. Should it be IFN-a?

Author Response

Reviewer 1

As the article aims to review current knowledge in glioma, the authors did good work in doing so. The authors have summarized the current knowledge thoroughly and systematically, and the manuscript is well written. This manuscript is suitable for publication, with a couple of text editing.

  1. Line 86: should it be N-cadherin instead of ’N cadherin ’?
  2. Figure 3, page 8: ‘INF-a’. Should it be IFN-a?

Thanks for your suggestions. All typographical errors you indicated have been fixed.

Reviewer 2 Report

“Tumor microenvironment in glioma invasion” by Tamai et al. is a comprehensive review that gives valuable insight into the role of the microenvironment in glioma invasion. Almost all aspects of the subject are covered. The manuscript is acceptable for publication after some minor considerations.

  1. I was intrigued by the last sentence in the Abstract: “…In addition, because the living body actively promotes tumor infiltration in response to the tumor, it is necessary to reconsider whether the invasion itself should be considered beneficial or not.” I have carefully reviewed the manuscript trying to find out how glioma invasion should be considered beneficial or not (for glioma spread into the brain parenchyma or for the novel therapeutic strategies?). I suggest rephrasing this sentence or trying to provide strong argumentation for this sentence in the manuscript.
  2. Page 2, Lines 68, 69: “Under normoxic conditions, prolyl hydroxylases hydroxylate two proline residues of HIF-1α, leading to proteasomal degradation of HIF-1α [21].” Please immediately add a sentence about what is happening with HIF-1α under hypoxic conditions.
  3. Figures 1, 2, 3, 4, and 6 need more detailed Figure Legends. The exception is Figure 5. Although the authors commented on Figures in the main manuscript’s main text, it would be easier to follow if there are short explanations in the Figure Legends. In addition, refer to Figures more than once in the main text.
  4. Page 4, Lines 114, 115: “GICs can be classified into three subtypes: proneural, mesenchymal, and classical.” The authors explained proneural, mesenchymal, but did not describe the classical GIC.
  5. Page 6, lines 171-176: “Furthermore, it has been reported that ECM promotes invasive ability by altering cellular signaling pathways in glioma cells, and the mechanical rigidity of ECM strongly regulates glioma cell proliferation and invasion [56]. The ECM is remodeled, and its rigidity is significantly increased compared to both juvenile and mature states during cancer genesis [50]. It has been reported that ECM stiffness may correlate with invasiveness and patient prognosis in adult gliomas [57].” This is somehow just hinted at without precise explanations.
  6. Figure 3 does not support the subtitle and the main text: “3.1.3. TAAs activate glioma invasion through the secretion of extracellular vesicles (EVs)”. It was said quite opposite that glioma cells shed EVs and influence astrocytes.
  7. “3.2.2. Roles of GAMs for changing microenvironments” and “3.2.3. Promotion of tumor invasion by GAMs”: These sections should be reordered. They sound upside-down.
  8. Page 11. Lines 402-405: “MMPs are a family of extracellular endopeptidases that selectively degrade components of the extracellular matrix. MMPs mediate the breakdown of the basal membrane and are implicated in tumor cell invasion. In addition, they appear to be important for the creation and maintenance of a microenvironment that facilitates tumor cell survival.” The authors have already explained MMPs. There is no need for repeating.
  9. Page 14, Lines 524, 525: “Some mechanisms alter the tumor microenvironment and activate invasiveness in each treatment.” This sentence should be removed. It interrupts the text flow.

Author Response

Reviewer 2

Comments and Suggestions for Authors

“Tumor microenvironment in glioma invasion” by Tamai et al. is a comprehensive review that gives valuable insight into the role of the microenvironment in glioma invasion. Almost all aspects of the subject are covered. The manuscript is acceptable for publication after some minor considerations.

I was intrigued by the last sentence in the Abstract: “…In addition, because the living body actively promotes tumor infiltration in response to the tumor, it is necessary to reconsider whether the invasion itself should be considered beneficial or not.” I have carefully reviewed the manuscript trying to find out how glioma invasion should be considered beneficial or not (for glioma spread into the brain parenchyma or for the novel therapeutic strategies?). I suggest rephrasing this sentence or trying to provide strong argumentation for this sentence in the manuscript.

According to the reviewer’s suggestion, the sentence “it is necessary to reconsider whether the invasion itself should be considered beneficial or not.” was added in the manuscript (page 18, lines 715-718).

Page 2, Lines 68, 69: “Under normoxic conditions, prolyl hydroxylases hydroxylate two proline residues of HIF-1α, leading to proteasomal degradation of HIF-1α [21].” Please immediately add a sentence about what is happening with HIF-1α under hypoxic conditions.

According to the reviewer’s suggestion, we added the sentence in page 2, line 71-72.

Figures 1, 2, 3, 4, and 6 need more detailed Figure Legends. The exception is Figure 5. Although the authors commented on Figures in the main manuscript’s main text, it would be easier to follow if there are short explanations in the Figure Legends. In addition, refer to Figures more than once in the main text.

We added detailed figure legends in figure 1, 2, 3, 4, and 6. Additionally, we referred to figures multiple times in the manuscript.

Page 4, Lines 114, 115: “GICs can be classified into three subtypes: proneural, mesenchymal, and classical.” The authors explained proneural, mesenchymal, but did not describe the classical GIC.

Thank for your suggestion. According to the reviewer’s suggestion, we revised manuscript as follows in page 4, line 118-122.

Each mechanism is associated with GIC subtypes. GICs can be classified into three subtypes: classical, mesenchymal, and proneural. Each subtype indicates different properties. The classical subtype contribute escaping apoptosis [39]. The mesenchymal subtype is associated with the stimulation of angiogenesis, and the proneural subtype subserves in invasion [40].

Page 6, lines 171-176: “Furthermore, it has been reported that ECM promotes invasive ability by altering cellular signaling pathways in glioma cells, and the mechanical rigidity of ECM strongly regulates glioma cell proliferation and invasion [56]. The ECM is remodeled, and its rigidity is significantly increased compared to both juvenile and mature states during cancer genesis [50]. It has been reported that ECM stiffness may correlate with invasiveness and patient prognosis in adult gliomas [57].” This is somehow just hinted at without precise explanations.

Thanks for your suggestion. As you suggested, this statement was ambiguous, we rewritten as follows in page 6, line 180-186.

These ECMs promote invasiveness of glioma cells by altering cellular signaling pathways. Another important factor relating to glioma invasion is rigidity. The mechanical rigidity of ECM positively regulates glioma cell proliferation and invasion [57]. The ECM is remodeled, and its rigidity is significantly increased compared to both juvenile and mature states during cancer genesis [51]. It has been reported that ECM stiffness correlates with glioma grading, and associate with tumor invasiveness and poor prognosis [58].

Figure 3 does not support the subtitle and the main text: “3.1.3. TAAs activate glioma invasion through the secretion of extracellular vesicles (EVs)”. It was said quite opposite that glioma cells shed EVs and influence astrocytes.

We changed the title 3. 1. 3. as follows.

The crosstalk between glioma cells and astrocytes via extracellular vesicles (EVs) contributes to glioma invasion

“3.2.2. Roles of GAMs for changing microenvironments” and “3.2.3. Promotion of tumor invasion by GAMs”: These sections should be reordered. They sound upside-down.

Thank for your suggestion. We reordered sections.

Page 11. Lines 402-405: “MMPs are a family of extracellular endopeptidases that selectively degrade components of the extracellular matrix. MMPs mediate the breakdown of the basal membrane and are implicated in tumor cell invasion. In addition, they appear to be important for the creation and maintenance of a microenvironment that facilitates tumor cell survival.” The authors have already explained MMPs. There is no need for repeating.

According to the reviewer’s comments, we deleted this sentence in the revised manuscript.

Page 14, Lines 524, 525: “Some mechanisms alter the tumor microenvironment and activate invasiveness in each treatment.” This sentence should be removed. It interrupts the text flow.

According to the reviewer’s comments, we deleted this sentence in the revised manuscript.

Reviewer 3 Report

Tamai et al., have generated a comprehensive review of the literature related to glioma invasion and subdivided the article into appropriate sections describing the mechanisms of glioma invasion. 

The article contributes to the field and is scientifically sound. Minor revisions can be made to the abstract to make it more attractive and flow smoothly. The rest of the article reads nicely.

Author Response

Thanks for your suggestions. The abstract has been changed as the reviewer suggested.